# OpenReview forum: "ASTPrompter: Weakly Supervised Automated Language Model Red-Teaming to Identify Low-Perplexity Toxic Prompts"
_ICML.cc/2025/Conference — Submitted to ICML 2025_

### Official Review · Reviewer_XXnJ · 2025-03-05

**Overall Recommendation:** 2

**Summary:**

This paper proposes ASTPrompter, an approach to automating the red-teaming of LLMs by generating harmful yet fluent (i.e., low-perplexity) prompts. While the underlying motivation is not new, and the vulnerability of LLMs to such attacks is well known in the community, our main contribution lies in modifying the reinforcement learning from human feedback (RLHF) framework. Specifically, we replace the conventional reward model (RM) with a combination of a toxicity RM and a perplexity-based criterion.

**Claims And Evidence:**

1. What's the main difference between ASTPrompter and [1]?
2. Intro: Pretraining Data Cleaning. The statement, "These models are trained on massive, minimally cleaned datasets primarily consisting of textual data scraped from the Internet," oversimplifies the data curation process. In reality, extensive efforts are made to clean and filter pretraining datasets, as documented in the technical reports of LLaMA 2 and LLaMA 3. These efforts include deduplication, content filtering, and rigorous quality control to mitigate biases and harmful content. The paper should acknowledge these preprocessing steps to provide a more accurate representation of modern LLM training pipelines.
3. Intro: "Empirically, these approaches result in prompts that are highly effective in triggering toxicity but are often nonsensical or unlikely to emerge during natural language model operation." I believe this is the difference between whether the user is adversarial or not.
4. Evaluation: Scientific Rigor. The description of the evaluation, "To do this, we build a dataset of rollouts from adversaries trained using our approach and then optimize a language model against them. We evaluate the model safety-tuned with this strategy and show a lower incidence of toxicity," lacks the necessary scientific rigor. The absence of quantitative results, such as toxicity reduction metrics, benchmark comparisons, or statistical significance tests, weakens the argument. A more thorough presentation of empirical results, including numerical evidence, would strengthen the paper’s claims and provide a clearer assessment of the method’s effectiveness.

[1] SmoothLLM: Defending Large Language Models Against Jailbreaking Attacks

**Essential References Not Discussed:**

The author missed relation to another lines of research on safety: Jailbreak attack and harmful finetuning attack

[1] Universal and Transferable Adversarial Attacks on Aligned Language Models
[2] Jailbreaking Black Box Large Language Models in Twenty Queries
[3] Jailbreak Attacks and Defenses Against Large Language Models: A Survey
[4] Vaccine: Perturbation-aware alignment for large language model aginst harmful fine-tuning
[5] Representation noising effectively prevents harmful fine-tuning on LLMs
[6] Buckle Up: Robustifying LLMs at Every Customization Stage via Data Curation
[7] Fine-tuning can cripple your foundation model; preserving features may be the solution
[8] Lazy safety alignment for large language models against harmful fine-tuning
[9] Navigating the safety landscape: Measuring risks in finetuning large language models
[10] Safe lora: the silver lining of reducing safety risks when fine-tuning large language models
[11] No two devils alike: Unveiling distinct mechanisms of fine-tuning attacks

**Experimental Designs Or Analyses:**

1. The only large SOTA LLM evaluated in the paper is llama 3.1-8b. More tests on various 8b, 13b llms are necessary to prove the effectiveness because we know tiny models like GPT-2 are vulnerable to attack as they are not fully aligned.
2. How would you compare your methods with gradient based attack like GCG? I'm aware that GCG is for harmfulness, but we can also optimize the model's output towards toxicity? Is your RLHF-based method stronger?
3. Is the toxicity score used in the training and evaluation the same model? If true, could it be because of reward hacking?
4. What is the implicit toxicity meantinoed in sec. 5.1? I found the definition of toxicity unclear in the paper, let alone implicit toxicity.
5. For black box attack, can you try testing on gpt-4o or claude? I assume this method may fail, but it's good to learn what the response looks like.
6. In sec. 5.2: what are the three dots: "Rewarding defender toxicity is necessary..."?
7. For the only 8b llama model, did you use the instruct or the pretrained one? If it's pretrained, it's also not safely aligned. Because the examples in the appendix reads like completing the text, and it sounds like the task for pretrained rather than aligned model.

**Methods And Evaluation Criteria:**

1. Lack of Explanation for Horizon=1 in Figure 1
The meaning of horizon=1 in Figure 1 is unclear, as there is no explanation provided in the caption. Clarifying this term would improve the reader’s understanding of the figure.

2. Ambiguity in Figure 1 (Left) as a Success Case
It is unclear why the left side of Figure 1 is considered a success. The blue text appears to be a factual statement rather than an explicitly toxic output. Further justification or a clearer example would help support this classification.

3. Measurement of Toxicity
The paper does not clearly define how toxicity is measured. Providing details on the toxicity evaluation method, such as the specific model or criteria used, would enhance clarity.

4. Relocating Background Sections
Sections 3.1.1 and 3.1.2 should be moved to the appendix, as they primarily cover background knowledge applicable to all RLHF methods. This would streamline the main text and improve readability.

5. Computation of Toxicity Score in Equation 2
The method for computing the toxicity score in Equation 2 is not explicitly stated. Is the score a scalar output from another LLM trained for toxicity detection? Clarifying this would help readers understand how the toxicity signal is integrated into the proposed method.

**Other Comments Or Suggestions:**

N/A

**Other Strengths And Weaknesses:**

See above

**Questions For Authors:**

See above

**Relation To Broader Scientific Literature:**

See above

**Theoretical Claims:**

1. Using IPO sounds correct as the toxicity model may gave incorrect score for the chosen and rejected pairs.
2. Math and Code in appendix LGTM.

---

> ### Author Rebuttal · Authors · 2025-03-31
>
> ## Ours vs. Robey, et al., 2023 vs. GCG
> Thank you for your feedback! [1] and our work have significant differences. First, we note that they are fundamentally different in their goals and methods. [1] is a defense method against attacks. We are an attack method.
> Concerning attacks mentioned in [1] as well as GCG:
> Prior work inserts random and programmatic perturbations into the prompt to search for failures, resulting in attacks that are neither high probability (which they then have to separately filter for using a perplexity filter) nor human-understandable.
> Prior work requires the hand-design of perturbations (e.g., insert, swap, patch, etc.), whereas ASTPrompter presents a fully differentiable optimization scheme that requires no human involvement
>
> [1] SmoothLLM: Defending Large Language Models Against Jailbreaking Attack
>
> ## Quantitative Results
> Results requested are already present in the work on Tables 1 and 2.
>
> We present quantitative attack success, attack sequence perplexity, toxicity induction metrics (showing our method increases the toxicity of a frozen LM), and benchmarks against baselines and other RL methods.
>
> ## Figure 1
> The upper-left figure represents that the attacker and defender had one chance to interact (through one continuation turn). We will modify our caption to clarify this.
> The output from the frozen LM in Figure 1, left, is not a factual statement [1], in particular not with the proportional quantifier “vast majority.” Based on available data ([1]), no single group commits such a proportion of terrorist attacks. Using an identity group to motivate a false statement represents attack success. [2] Furthermore, we note that LLMs have been shown to exhibit harmful bias in which groups they associate with violence [2].
> - [1] https://www.visionofhumanity.org/maps/global-terrorism-index/#/
> - [2] Large Language Models Associate Muslims with Violence (Abid, et al., 2021)
>
> ## More Architectures
> Since, unlike other models at comparable scales, LLaMA models have extensive and documented pre-training safety mitigations, we strongly believe these results are representative of worst-case attack performance for models of the same size.
>
> ## Reward Hacking
> To mitigate reward hacking, we observe that our attacks produce fluent and coherent outputs (see Appendices D, E, and G) and test black-box approaches.
> Additionally, we include zero-shot evaluations against alternative common toxicity scores below, further mitigating concerns about reward hacking. Against LlamaGuard3 (toxicity score is 0-1 normalized P(“unsafe”|prompt) by Llamaguard), for llama-3.1-8b attacker trained against Detoxify, we have
>
> | | Defender Toxicity |
> |-|-|
> |Ours|0.075|
> |Baseline|0.023|
>
> ## Measuring and Defining Toxicity
> As stated in the paper (Sections 3.1.4, Section 4.1), we measure toxicity using Detoxify.
> As stated in our paper (lines 033-034, right), we use the widely accepted definition of toxicity developed by Perspective API [1] [2]. We define implicit toxicity as text using coded or indirect language to be rude, disrespectful, unreasonable, or otherwise likely to cause a user to leave a conversation. [4][5]
> - [1] Measuring and Mitigating Unintended Bias in Text Classification. (Dixon, et al., 2018)
> - [2] A new generation of perspective api: Efficient multilingual character-level transformers. (Lees, 2022).
> - [4] Latent hatred: A benchmark for understanding implicit hate speech. (ElSherief, et. al., 2021)
> - [5] Unveiling the implicit toxicity in large language models. (Wen, et. al., 2023)
>
> ## Commercial Models
> In fact, our method succeeded in increasing toxicity against Claude 3.5 Sonnet (cutoff 20241022).
>
> In particular, we include here black-box attack results against Claude 3.5 Sonnet 20241022, using our Llama-3.1-8B attacker trained against the Detoxify metric. We score toxicity in this experiment using two classifiers, Detoxify as used in our work, as well as against LlamaGuard3, a state of the art llm-as-judge detoxification model.
>
> | |Baseline|Ours|
> |-|-|-|
> |Detoxify|0.981%|1.999%|
> |Llamaguard|0.336%|2.239%|
>
> We achieve a 6.668 times increase in toxicity using the toxicity metric we trained against (Detoxify), and even generalized to 2.03 times increase in toxicity zero-shot to a new toxicity metric—achieving fully black-box attack success.
>
> ## Pretraining vs IFT
> We used the pretrained model. Uniquely, the pretrained LLaMA model has extensive safety mitigations with data filtering, content filtering, and quality control. Indeed this is not true of other pretrained models.
>
> ## Extra References
> Thank you! We will add these references to our related work. In comparison to this list, our work
>
> - [1] Provides low-perplexity sequences and doesn’t rely on high-perplexity adversarial suffixes.
> - [2] Is effective in a black-box setting without on-policy sampling
> - [3-11] Notably, we are primarily focused on attacks and not mitigations in our work.
>
> We are unsure how reference [7] is relevant.

---

> > ### Comment · Reviewer_XXnJ · 2025-04-04
> >
> > I agree with other reviewers that the rebuttal can help consolidate the paper into a stronger version for the next submission.

---

### Official Review · Reviewer_hv13 · 2025-03-10

**Overall Recommendation:** 2

**Summary:**

This paper introduces ASTPrompter, a Reinforcement Learning (RL) based approach for automated red-teaming of Large Language Models (LLMs). The method is designed to identify prompts that elicit toxic outputs from a defender LLM while also maintaining low perplexity, ensuring the generated prompts are likely to occur naturally. ASTPrompter formulates red-teaming as an Adaptive Stress Testing (AST) problem and solves it using an online and weakly supervised Identity Preference Optimization (IPO) scheme. The authors evaluate their approach against various baseline models and ablation studies, demonstrating improved toxicity elicitation rates and maintained prompt likelihood across different model scales. They also explore the downstream utility of their method for toxicity mitigation.

**Claims And Evidence:**

The paper claims that ASTPrompter effectively identifies low-perplexity prompts that elicit toxicity from LLMs, outperforming baselines and maintaining prompt likelihood. While the experimental results in Table 1 and Figure 3 show that ASTPrompter achieves higher toxicity rates compared to baselines and ablations, the claim of novelty is questionable given existing RL-based red-teaming approaches such as [1]. The claim that IPO converges faster than PPO is made without direct empirical evidence comparing training time or efficiency. The claim of improved safety through downstream detoxification is supported by initial experiments in Table 3, but further evidence and analysis could strengthen this claim.

[1] Chen, Xuan, et al. "When LLM Meets DRL: Advancing Jailbreaking Efficiency via DRL-guided Search." NeurIPS, 2024.

**Essential References Not Discussed:**

[1] also utilizes RL for jailbreaking LLMs and is directly relevant to the proposed approach. Failing to cite and compare against this recent and highly relevant work weakens the novelty claim and contextualization of ASTPrompter.

[1] Chen, Xuan, et al. "When LLM Meets DRL: Advancing Jailbreaking Efficiency via DRL-guided Search." NeurIPS, 2024.

**Experimental Designs Or Analyses:**

The experimental design includes comparisons against baselines and ablation studies to evaluate the contribution of different reward terms and weak supervision. Both white-box and black-box attack scenarios are considered. The use of Convokit Reddit corpus as non-toxic prompts and RealToxicityPrompts for weak supervision is described. However, the paper lacks a direct empirical comparison with other existing red-teaming methods, limiting the assessment of ASTPrompter's relative performance. The choice of outdated attack models (GPT-2, GPT-2 XL) raises concerns about the relevance of the evaluation in the context of rapidly evolving LLMs. The lack of time efficiency comparisons for IPO vs. PPO weakens the claim of faster convergence for IPO.

**Methods And Evaluation Criteria:**

The proposed method uses a RL framework with IPO for optimizing an adversary policy to generate toxic prompts. The reward function incorporates defender toxicity, combined toxicity, and prompt perplexity. Weak supervision using RealToxicityPrompts is introduced to improve convergence. The evaluation criteria include prompt perplexity, defense toxicity, and combined toxicity, measured using the Detoxify model. While these metrics are relevant for the task, the choice of Detoxify as the toxicity evaluation model is questionable as it may not represent state-of-the-art toxicity detection and has known biases. The evaluation environments are limited to GPT-2, GPT-2 XL, TinyLlama, and Llama-3.1-8b, and could benefit from including more advanced and recent LLMs.

**Other Comments Or Suggestions:**

N/A

**Other Strengths And Weaknesses:**

Strengths:

- The formulation of red-teaming as an Adaptive Stress Testing problem is interesting.

- The ablation studies provide insights into the contribution of different reward terms.

- The exploration of downstream detoxification is a promising direction.

Weaknesses:

- Lack of Novelty: The use of RL for LLM red-teaming is not entirely novel, and the paper fails to adequately compare with related work like [1].

- Missing Empirical Evidence: The claim of faster IPO convergence compared to PPO is not empirically supported.

- Outdated Attack Models: The evaluation primarily uses older models like GPT-2 and GPT-2 XL; more recent and advanced LLMs should be included.

- Questionable Toxicity Metric: The use of Detoxify (2020) as the toxicity judge is outdated and may not accurately reflect state-of-the-art toxicity detection.

- Lack of Empirical Comparison: The paper fails to empirically compare ASTPrompter against other existing automated red-teaming methods.

[1] Chen, Xuan, et al. "When LLM Meets DRL: Advancing Jailbreaking Efficiency via DRL-guided Search." NeurIPS, 2024.

**Questions For Authors:**

See the weaknesses.

**Relation To Broader Scientific Literature:**

The paper relates to the growing body of literature on automated red-teaming of LLMs and AI safety. It builds upon previous work using RL for red-teaming and incorporates techniques like IPO and weak supervision. However, the paper fails to adequately acknowledge and compare against closely related work, particularly the RL-based jailbreaking approach such as [1]. The positioning within the broader literature could be strengthened by explicitly discussing and contrasting ASTPrompter with other existing automated red-teaming methods and frameworks.

[1] Chen, Xuan, et al. "When LLM Meets DRL: Advancing Jailbreaking Efficiency via DRL-guided Search." NeurIPS, 2024.

**Theoretical Claims:**

There are no explicit theoretical claims or proofs presented in the paper. The method is primarily empirically driven and focused on algorithmic design and evaluation.

---

> ### Author Rebuttal · Authors · 2025-03-31
>
> ## Other Models
>
> In addition to GPT 2 and GPT2-XL, we already report results using Llama-8b models and TinyLlama in the article (Table 1), demonstrating successful attacks with low perplexity prompts.
>
> Furthermore, we include here **black-box attack results against Claude 3.5 Sonnet (cutoff 20241022)**, using our Llama-3.1-8B attacker trained against the Detoxify metric. We score toxicity in this experiment using two classifiers: Detoxify, as used in our work, and LlamaGuard3, a state-of-the-art llm-as-judge detoxification model. Toxicity is measured by 0-1 normalized P(“unsafe”|prompt) given by the LlamaGuard model. We use prompts from [1] as our baseline.
>
> | |Baseline|Ours|
> |-|-|-|
> |Detoxify|0.981%|1.999%|
> |Llamaguard|0.336%|2.239%|
>
> We achieve a 6.668 times increase in toxicity using the toxicity metric we trained against (Detoxify). Additionally, we even generalize to 2.03 times increase in toxicity zero-shot with a new toxicity metric, achieving full black-box attack success.
>
> [1] Bot-Adversarial Dialogue for Safe Conversational Agents (Xu et al., 2021)
>
> ## Optimization Method
>
> Our article does not make the claim that IPO converges more stably than PPO, since this has already been shown by [1][2]. Although the optimization method is only a means to validate our formulation and training scheme, we mention in our work that DPO, when available, converges more stably than PPO, citing the results of the DPO authors [1]. However, DPO, unlike IPO and PPO, requires a rational ranking. Hence, IPO was chosen.
>
> - [1] Direct Preference Optimization: Your Language Model is Secretly a Reward Model (Rafailov, et al., 2023)
> - [2] A General Theoretical Paradigm to Understand Learning from Human Preferences (Azar et al., 2023)
>
> ## Baseline RL for LLM vs. AST
> We do not claim that the use of RL for LLM red-teaming is our paper’s primary novelty. Instead, what is novel in our work is the multi-objective reward function that encourages lower perplexity attacks. Through AST, our key insight is that likelihood should be considered and optimized to avoid reward hacking that produces failures under unlikely circumstances.
>
> ## Chen et al., 2024
> We note that the reward function of [1] does not account for likelihood and instead optimizes for the closeness between the target model’s generation and a reference answer to a harmful question. Such approaches reduce the diversity of the outputs and attacks (to be semantically similar to the target output), which is undesirable in red-teaming use cases since such optimizations will narrow down to one specific type of attack which then can be defended against with carefully crafted heuristics. [2]
>
> - [1] When LLM Meets DRL: Advancing Jailbreaking Efficiency via DRL-guided Search. (Chen, et. al., 2024).
> - [2] Curiosity-driven Red-teaming for Large Language Models (Hong, et. al., 2024)
>
> ## Toxicity Metric
>
> We choose Detoxify because it is a locally-runnable model with good representation in literature as a heuristic for unwanted toxic data in pretraining (Henderson et al., NIPS 2022), assistants (Köpf et al., NIPS 2023), to evaluate detoxification success (Korbak et al., ICML 2023) and others.
>
> Unlike Detoxify, which is a locally runnable model, Perspective API and OpenAI API—both online APIs commonly used for measuring toxicity—have significant rate limits that render them unsuitable for being called in the training loop.
> Here, we present evaluations of our approach using Llama-3-8B and zero-shot generalization against a novel, locally runnable, toxicity metric: LlamaGuard3. Against LlamaGuard3, a state-of-the-art llm-as-judge detoxification model, we obtain the following black-box attack success for a llama-3.1-8b adversary trained to elicit toxicity (measured by Detoxify) from a llama-3.1-8b defender. The toxicity score in these results is measured as the 0-1 normalized P(“unsafe”|prompt) given by the LlamaGuard model.
>
> | | Defender Toxicity |
> |-|-|
> |Ours|0.075|
> |Baseline|0.023|
>
> Compared to the baseline (unturned llama-3.1-8b), we achieve 3.5 times higher incidence of toxicity. We note that our main contribution is to demonstrate that optimizing our formulation gives low-perplexity red-teaming attacks. We would expect future work to adapt the toxicity model to be application-specific.
>
> ## Empirical Comparisons
> We already provide direct empirical comparisons of our method to several popular automated approaches. These results are located in Table 1 and Table 2.
>
> In particular, we compare our approach to several other gradient-based methods: supervised fine-tuning on human-written prompts intended to elicit toxicity (Table 1) and reinforcement-learning-based red-teaming without weak supervision and a perplexity reward (Table 2, RL baseline)—which constitutes an adapted version of the reinforcement learning driven method proposed by Perez et. al. Furthermore, we test against non-gradient-based automated red-teaming methods including human-written attack prompts.

---

> > ### Comment · Reviewer_hv13 · 2025-04-02
> >
> > Thank you for your response. I still have a few concerns regarding your rebuttal.
> >
> > ### Other Models
> >
> > If I understand correctly, the reported numbers represent  P(“unsafe”|prompt) given by the LlamaGuard model. I don't think the performance of the proposed method is very impressive given these numbers.
> >
> > ### Optimization Method
> >
> > I also read the authors' rebuttal to other reviewers. How do you validate your claim that " DPO, when available, converges more stably than PPO" and "DPO, unlike IPO and PPO, requires a rational ranking"? Additionally, as the authors claim that the main contribution of this work is the multi-objective reward design, I would like to bring [1, 2] into the authors' attention where both PPO and DPO could be extended to the multi-objective scenario. I would appreciate it if the authors could provide further explanation over why chose IPO.
> >
> > [1] Kaiwen Li, Tao Zhang, and Rui Wang. Deep reinforcement learning for multiobjective optimization. IEEE transactions on cybernetics, 2020.
> >
> > [2] Zhanhui Zhou, Jie Liu, Jing Shao, Xiangyu Yue, Chao Yang, Wanli Ouyang, and Yu Qiao. Beyond one-preference-fits-all alignment: Multi-objective direct preference optimization. In Findings of ACL, 2024.
> >
> > ### Baseline RL for LLM vs. AST
> >
> > The authors acknowledge that the main contribution of this work is the reward design, which is a common practice in RL work. Simply proposing a customized reward limits the work's novelty and makes it far from the bar of ICML.
> >
> > ### Empirical Comparisons
> >
> > There exist many more recent work regarding auto red-teaming (e.g., GPTFuzz, Chen et al. (2024)). I would be happy to see how these more recent methods perform against your method.

---

> > > ### Author Response · Authors · 2025-04-02
> > >
> > > We thank you for your timely feedback. We would like to clarify a few points:
> > >
> > > ## Contributions and Reward Design
> > > It is certainly true that RL research often hinges on how objectives are specified. However, we believe our paper goes beyond proposing a custom reward, offering contributions that are relevant to the ICML community:
> > > - Formulation as AST: We introduce a conceptual framework casting automated red-teaming as an Adaptive Stress Testing problem, which focuses on likely text failures. This distinguishes our work from "jailbreaking" approaches that rely on unnatural, highly adversarial prompts; such prompts are rarely produced by typical users. By contrast, our AST framing emphasizes failure scenarios that might naturally arise in day-to-day interactions.
> > > - Weakly Supervised, Online Preference Method: We combine a multi-objective reward design with online sampling, pairwise preference training, and weak supervision to explore the prompt space. Approaches that simply maximize toxicity risk producing highly unnatural prompts. In our setting, we also preserve low perplexity, so the model's outputs remain probable under the frozen LM—yet still induce undesired toxic responses.
> > > - Empirical Evidence of "Likely Toxicity": Our results confirm that traditional automated red-teaming can degrade prompt likelihood drastically. By including a perplexity/language-model-likelihood term, we preserve realistic prompts—highlighting how "ordinary" user prompts may still lead to toxic outputs. This issue is highly relevant to real-world LLM deployments.
> > >
> > > We would like to observe that, in general, much of machine learning can be viewed as either changing the modeling technique or objective function of an optimization problem. We contribute to the latter case.
> > >
> > > ## Choice of Optimization Methods (IPO vs. DPO vs. PPO vs. Others)
> > > Our primary focus is on the novel formulation itself and the effects of optimizing it. We discuss IPO, DPO, and PPO because they are widely used in the LLM post-training domain, well-understood, and represent typical candidates. The specific choice of IPO over DPO or PPO is supported by both the literature cited in our paper and the discussion in the "Optimization Method" section of the rebuttal. Nevertheless, any RL technique that can handle multi-objective and rational-ranking constraints could, in principle, solve our formulation.
> > >
> > > ## GPTFuzz and Chen et al.
> > > These methods largely represent jailbreaking strategies, in which prompts are high perplexity and require an adversarial mindset from the user. By design, such prompts differ from the more "likely" prompts we target which is by definition more likely to arise during autoregression. Indeed, in our experiments with a reinforcement-learning baseline (Table 2, "RL baseline" referencing Perez et al.), we show that removing perplexity-related constraints significantly increases perplexity while eliciting toxicity. Because Chen et al. and similar techniques rely on prompts that would be improbable in normal conversation, they tackle an important but distinct scenario (adversarial "jailbreaking") from the one motivating our AST perspective (high-probability failures).
> > >
> > > Furthermore, approaches like GPTFuzz require the hand-design of base mutations, whereas our approach requires no human involvement at all during the attack process.

---

### Official Review · Reviewer_rrN8 · 2025-03-14

**Overall Recommendation:** 2

**Summary:**

The paper presents a reinforcement learning-based red-teaming method to identify prompts that elicit toxic outputs from language models while maintaining fluency. The approach uses Adaptive Stress Testing (AST) and Identity Preference Optimization (IPO) to generate prompts with high likelihood and increased toxicity. The method outperforms baselines approaches.

**Claims And Evidence:**

The paper claims ASTPrompter produces more effective red-teaming prompts than baselines, the paper presents empirical results (Table 1) showing that ASTPrompter elicits higher toxicity from defender models than other approaches, such as human-written attacks (BAD) and fine-tuned models (SFT). The toxicity rate is up to 23 times higher than baselines while maintaining low perplexity.

The paper claims ASTPrompter remains effective in both white-box and black-box attack settings, and presents cross-model evaluation results, shows the proposed approach attack unseen defender models from different families (e.g., GPT-2 attacking Llama-3.1-8b). The approach remains effective, generating 5.4–14x increased toxicity in black-box settings.

**Essential References Not Discussed:**

n/a

**Experimental Designs Or Analyses:**

The paper evaluates ASTPrompter’s effectiveness using multiple defender models (e.g., GPT-2, GPT-2 XL, TinyLlama, Llama-3.1-8b), and multiple adversary models (e.g., white & black box) and for multiple training objectives, where experiment varies how the reward function weights toxicity elicitation and prompt likelihood (perplexity).

**Methods And Evaluation Criteria:**

The paper effectively uses AST and IPO for red-teaming, optimizing prompts for both toxicity elicitation and low perplexity. Evaluation metrics, including defender toxicity and combined toxicity, ensure fluency and effectiveness.

**Other Comments Or Suggestions:**

N/A

**Other Strengths And Weaknesses:**

N/A

**Questions For Authors:**

the paper mainly uses IPO as the optimization method, how does the proposed method perform with other approaches such as DPO, RLHF, which are more classic?

**Relation To Broader Scientific Literature:**

The paper contributes to several areas such as automated red-teaming, adversarial prompting, reinforcement learning, LLM, and trustworthy LLM.

**Theoretical Claims:**

the paper does not include any theoretical proofs

---

> ### Author Rebuttal · Authors · 2025-03-31
>
> Thank you for your review! With respect to IPO vs. DPO vs. RLHF-PPO type methods, we do not compare our method to DPO, since we use a multi-objective reward function and DPO assumes rationally ranked responses. Since the LM perplexity/toxicity evaluations may not be exactly rationally ranked, the DPO formulation is less appropriate.
>
> RLHF generally includes a broad range of optimization methods applied to rewards/data from human feedback; however, it commonly uses PPO as the optimization method. We note that DPO converges more stably than PPO [1] [3]  and that our primary contribution involves the effects of optimizing our novel formulations rather than the optimization method itself. In order to achieve stability in PPO, one has to make many algorithmic changes that are out of the scope of this work [2]. For this reason, many notable models (e.g., the Llama 3 family [4]) are trained with DPO rather than PPO.
>
> For the camera-ready draft, we will additionally use PPO to optimize our formulation to highlight its efficacy and illustrate the advantages of the preference-learning approach.
>
> - [1] Direct Preference Optimization: Your Language Model is Secretly a Reward Model (Rafailov et al., 2024)
> - [2] Unpacking DPO and PPO: Disentangling Best Practices for Learning from Preference Feedback (Ivison et al., 2024).
> - [3] A General Theoretical Paradigm to Understand Learning from Human Preferences (Azar et al., 2023)
> - [4] The Llama 3 Herd or Models (Grattafiori et al., 2024)

---

### Official Review · Reviewer_LWJw · 2025-03-20

**Overall Recommendation:** 2

**Summary:**

The paper introduces ASTPrompter, a reinforcement learning-based red-teaming approach that uses Adaptive Stress Testing and online weakly supervised Identity Preference Optimization to find toxic prompts. The method outperforms baselines by eliciting 2-23X more toxicity while maintaining fluency, and works in both white-box and black-box attack settings. The generated adversarial prompts can serve as negative training samples to improve LLM safety tuning.

**Claims And Evidence:**

The claims made in the submission are generally well-supported by empirical results.

**Essential References Not Discussed:**

N/A

**Experimental Designs Or Analyses:**

The paper’s experiments are generally well-structured. However, the study assumes that low perplexity equates to a natural prompt, which has not been directly tested. Additional metrics should be included to evaluate whether the prompts are truly natural.

**Methods And Evaluation Criteria:**

The proposed methods and evaluation criteria in the paper are well-suited for the problem of automated red-teaming and identifying likely toxic prompts.

**Other Comments Or Suggestions:**

N/A

**Other Strengths And Weaknesses:**

**Strengths:**

- Introduces a novel AST-based formulation for LLM red-teaming. AST is commonly used in safety-critical fields but has not been applied to language model security testing before. Using AST for red-teaming can generate more natural toxic prompts, which can be leveraged to fine-tune models for enhanced safety.
- Effective in black-box settings, making it a practical attack method.


**Weaknesses:**

- The paper claims IPO is superior to PPO for multi-objective optimization but does not provide experimental comparisons between the two methods. PPO could also be used for multi-objective optimization.

- The paper asserts that gradient-based red-teaming methods generate unnatural prompts but does not offer direct empirical comparisons of perplexity or fluency.

- While ASTPrompter generates low-perplexity toxic prompts, it is unclear whether these prompts resemble real-world adversarial prompts encountered in deployed LLMs.

**Questions For Authors:**

Did you evaluate ASTPrompter’s ability to elicit toxicity using multiple toxicity classifiers, such as the Perspective API or OpenAI API?

Is there a comparison of perplexity or fluency between ASTPrompter and gradient-based methods?

**Relation To Broader Scientific Literature:**

The paper discusses red-teaming, jailbreaking, and toxicity attacks. Its contributions are valuable to both the research community and model owners, helping to reassess red-teaming approaches. This work addresses a gap in current research and offers solutions for improving red-teaming strategies.

**Theoretical Claims:**

N/A, the paper mainly relies on empirical results.

---

> ### Author Rebuttal · Authors · 2025-03-31
>
> ## Fluency and Naturalness
>
> We do not directly test the relationship between low perplexity and naturalness, as the goal of our paper is to identify low perplexity prompts, independent of their naturalness. However, we do qualitatively observe a relationship, as shown in the rollouts in Appendices F and G.
>
> As model scales increase, evidence shows that the LM test-loss (i.e. log-perplexity) of natural text decreases [2]—which means that maximum-likelihood sampling will generate relatively more natural text that is within distribution of the test corpus. However, red-teaming attacks typically have high perplexity since they are slightly out of distribution. For instance, red-teaming approaches such as BAD [1], a dataset of human-written prompts, have high naturalness as they are created by humans, while also having high perplexity according to the language model.
>
> Hence, in our work we aim to optimize both the likelihood of the prompts and the amount of toxicity they induce. Naturalness is a consequence of optimizing likelihood, but not a goal of it. Instead, we focus on prompts that are likely to be generated during autoregression, regardless of their naturalness.
>
> - [1] Bot-Adversarial Dialogue for Safe Conversational Agents (Xu et al., 2021)
> - [2] Scaling Laws for Neural Language Models (Kaplan et al., 2020)
>
> ## IPO vs PPO
>
> Notably, we did not claim in the paper that IPO is superior to PPO for multiple-objective optimization.
>
> We argue in our work that DPO, when available, converges more stably in the paper than PPO. This is shown by the DPO authors [1]. With respect to multi-objective optimization, we were describing this as a limitation of DPO not PPO. That is, it is DPO that, unlike IPO and PPO, requires a rational ranking. We will revise the language in the article to clarify this.
>
> [1] Direct Preference Optimization: Your Language Model is Secretly a Reward Model (Rafailov, et al., 2023)
>
> ## Empirical Perplexity Comparison
>
> We provide the requested direct empirical comparisons of perplexity to several popular gradient-based approaches in Table 1 and Table 2 in the article.
>
> In particular, we compare our approach to the following gradient-based methods: supervised fine-tuning on human-written prompts intended to elicit toxicity (Table 1) and reinforcement-learning-based red-teaming without weak supervision or a perplexity reward (Table 2, RL baseline)—an adapted version of the reinforcement learning driven method proposed by Perez et. al.
>
> Although these results demonstrate that our method achieves the lowest attack perplexity compared to related works, it does not directly consider fluency. However, perplexity defines the inverse surprise of a phrase occurring, which is correlated with acceptability (i.e., fluency) of text for humans; this has been long-established in cognitive science literature [1] [2] [3] [4]. As perplexity is a measure of generation likelihood, we argue that low-perplexity prompts, no matter how fluent, are more likely to occur during standard autoregression.
>
> - [1] The effect of word predictability on reading time is logarithmic (Smith, et. al., 2013)
> - [2] Expectation-based syntactic comprehension (Levy, et. al., 2007)
> - [3] Data from eye-tracking corpora as evidence for theories of syntactic processing complexity (Demberg, et al., 2008)
> - [4] A Probabilistic Earley Parser as a Psycholinguistic Model (Hale, 2001)
>
> ## Real-World Fluency
> Though the prompts generated are typically fluent, we do not claim that these prompts resemble real-world human adversaries. However, we propose that they are more likely to arise from autoregression of a frozen LLM as they are lower in perplexity as scored by that model. Furthermore, we demonstrate that an LLM trained against the prompts generated by our approach is less toxic against a variety of attack strategies (Section 6).
>
> ## Other Classifiers
> Both the perspective API and OpenAI API have significant rate limits that render them unsuitable for being called in the training loop. However, we tested another local, LLM based toxicity classifier—LlamaGuard—and provide results below.
> Against LlamaGuard3, a state-of-the-art llm-as-judge detoxification model, we obtain the following black-box attack success for a llama-3.1-8b adversary trained with our method to elicit toxicity (measured by Detoxify) from a llama-3.1-8b defender.
> Toxicity score here is measured by the 0-1 normalized P(“unsafe”|prompt) given by LlamaGuard.
>
> | | Defender Toxicity |
> |-|-|
> |Ours|0.075|
> |Baseline|0.023|
>
> In comparison to untuned baseline, we achieve 3.5 times higher incidence of toxicity.
>
> ## Comparison to Other Gradient-Based Methods
> The paper provides that comparison in Table 1 and Table 2.
>
> There is a perplexity comparison provided to supervised fine-tuning on human-written prompts intended to elicit toxicity (Table 1) and reinforcement-learning-based red-teaming without weak supervision and a perplexity reward (Table 2, RL baseline).

---

### Decision · Program_Chairs · 2025-05-01

**Decision:**

Reject

**Comment:**

The paper introduces ASTPrompter, a reinforcement learning-based red-teaming approach that uses Adaptive Stress Testing and online weakly supervised Identity Preference Optimization to find toxic prompts. After rebuttal, all reviewers still have concerns about the novelty and experimental results with baselines. Authors did not fully addressed the concerns from reviewers after rebuttal. AC reads all comments and agrees with reviewers' recommendation. Thus, AC tends to reject this paper.